# An Empirical Study on The Properties of Random Bases for Kernel Methods

**Maximilian Alber, Pieter-Jan Kindermans, Kristof T. Schütt**
Technische Universität Berlin
maximilian.alber@tu-berlin.de

**Klaus-Robert Müller**
Technische Universität Berlin
Korea University
Max Planck Institut für Informatik

**Fei Sha**
University of Southern California
feisha@usc.edu

## Abstract

Kernel machines as well as neural networks possess universal function approximation properties. Nevertheless in practice their ways of choosing the appropriate function class differ. Specifically neural networks learn a representation by adapting their basis functions to the data and the task at hand, while kernel methods typically use a basis that is not adapted during training. In this work, we contrast random features of approximated kernel machines with learned features of neural networks. Our analysis reveals how these random and adaptive basis functions affect the quality of learning. Furthermore, we present basis adaptation schemes that allow for a more compact representation, while retaining the generalization properties of kernel machines.

## 1 Introduction

Recent work on scaling kernel methods using random basis functions has shown that their performance on challenging tasks such as speech recognition can match closely those by deep neural networks [22, 6, 35]. However, research also highlighted two disadvantages of random basis functions. First, a large number of basis functions, i.e., features, are needed to obtain useful representations of the data. In a recent empirical study [22], a kernel machine matching the performance of a deep neural network required a much larger number of parameters. Second, a finite number of random basis functions lead to an inferior kernel approximation error that is *data-specific* [30, 32, 36].

Deep neural networks learn representations that are adapted to the data using end-to-end training. Kernel methods on the other hand can only achieve this by selecting the optimal kernels to represent the data – a challenge that persistently remains. Furthermore, there are interesting cases in which learning with deep architectures is advantageous, as they require exponentially fewer examples [25]. Yet arguably both paradigms have the same modeling power as the number of training examples goes to infinity. Moreover, empirical studies suggest that for real-world applications the advantage of one method over the other is somewhat limited [22, 6, 35, 37].

Understanding the differences between approximated kernel methods and neural networks is crucial to use them optimally in practice. In particular, there are two aspects that require investigation: (1) How much performance is lost due to the kernel approximation error of the random basis? (2) What is the possible gain of adapting the features to the task at hand? Since these effects are expected to be data-dependent, we argue that an empirical study is needed to complement the existing theoretical contributions [30, 36, 20, 32, 8].

In this work, we investigate these issues by making use of the fact that approximated kernel methods can be cast as shallow, one-hidden-layer neural networks. The bottom layers of these networks are random basis functions that are generated in a *data-agnostic* manner and are *not adapted* during training [30, 31, 20, 8]. This stands in stark contrast to, even the conventional single layer, neural network where the bottom-layer parameters are optimized with respect to the data distribution and the loss function. Specifically, we designed our experiments to distinguish four cases:

- **Random Basis (RB):** we use the (approximated) kernel machine in its traditional formulation [30, 8].

- **Unsupervised Adapted Basis (UAB):** we adapt the basis functions to better approximate the true kernel function.

- **Supervised Adapted Basis (SAB):** we adapt the basis functions using kernel target alignment [5] to incorporate label information.

- **Discriminatively Adapted Basis (DAB):** we adapt the basis functions with a discriminative loss function, i.e., optimize jointly over basis and classifier parameters. This corresponds to conventional *neural network* optimization.

These experiments allow us to isolate the effect of the randomness of the basis and contrast it to data- and task-dependent adaptations. We found that adapted bases consistently outperform random ones: an unsupervised basis adaption leads to a better kernel approximation than a random approximation, and, when considering the task at hand, a supervised kernel basis leads to a even more compact model while showing a superior performance compared to the task-agnostic bases. Remarkably, this performance is retained after transferring the basis to another task and makes this adaption scheme a viable alternative to a discriminatively adapted basis.

The remainder is structured as follows. After a presentation of related work we explain approximated kernel machines in context of neural networks and describe our propositions in Sec. 3. In Sec. 4 we quantify the benefit of adapted basis function in contrast to their random counterparts empirically. Finally, we conclude in Sec. 5.

## 2 Related work

To overcome the limitations of kernel learning, several approximation methods have been proposed. In addition to Nyström methods [34, 7], random Fourier features [30, 31] have gained a lot of attention. Random features or (faster) enhancements [20, 9, 39, 8] were successfully applied in many applications [6, 22, 14, 35], and were theoretically analyzed [36, 32]. They inspired scalable approaches to learn kernels with Gaussian processes [35, 38, 23]. Notably, [2, 24] explore kernels in the context of neural networks, and, in the field of RBF-networks, basis functions were adapted to the data by [26, 27].

Our work contributes in several ways: we view kernel machines from a neural network perspective and delineate the influence of different adaptation schemes. None of the above does this. The related work [36] compares the data-dependent Nyström approximation to random features. While our approach generalizes to structured matrices, i.e., fast kernel machines, Nyström does not. Most similar to our work is [37]. They interpret the Fastfood kernel approximation as a neural network. Their aim is to reduce the number of parameters in a convolutional neural network.

## 3 Methods

In this section we will detail the relation between kernel approximations with random basis functions and neural networks. Then, we discuss the different approaches to adapt the basis in order to perform our analysis.

### 3.1 Casting kernel approximations as shallow, random neural networks

Kernels are pairwise similarity functions $k(x, x') : \mathbb{R}^d \times \mathbb{R}^d \mapsto \mathbb{R}$ between two data points $x, x' \in \mathbb{R}^d$. They are equivalent to the inner-products in an intermediate, potentially infinite-dimensional feature

space produced by a function $\phi : \mathbb{R}^d \mapsto \mathbb{R}^D$

$$k(x, x') = \phi(x)^T \phi(x') \tag{1}$$

Non-linear kernel machines typically avoid using $\phi$ explicitly by applying the kernel trick. They work in the dual space with the (Gram) kernel matrix. This imposes a quadratic dependence on the number of samples $n$ and prevents its application in large scale settings. Several methods have been proposed to overcome this limitation by approximating a kernel machine with the following functional form

$$f(x) = W^T \hat{\phi}(x) + b, \tag{2}$$

where $\hat{\phi}(x)$ is the approximated kernel feature map. Now, we will explain how to obtain this approximation for the Gaussian and the ArcCos kernel [2]. We chose the Gaussian kernel because it is the default choice for many tasks. On the other hand, the ArcCos kernel yields an approximation consisting of rectified, piece-wise linear units (ReLU) as used in deep learning [28, 11, 19].

**Gaussian kernel**    To obtain the approximation of the Gaussian kernel, we use the following property [30]. Given a smooth, shift-invariant kernel $k(x - x') = k(z)$ with Fourier transform $p(w)$, then:

$$k(z) = \int_{R^d} p(w) e^{j w^T z} dw. \tag{3}$$

Using the Gaussian distribution $p(w) = \mathcal{N}(0, \sigma^{-1})$, we obtain the Gaussian kernel

$$k(z) = \exp^{-\frac{\|z\|_2^2}{2\sigma^2}}.$$

Thus, the kernel value $k(x, x')$ can be approximated by the inner product between $\hat{\phi}(x)$ and $\hat{\phi}(x')$, where $\hat{\phi}$ is defined as

$$\hat{\phi}(x) = \sqrt{\frac{1}{D}} [\sin(W_B^T x), \cos(W_B^T x)] \tag{4}$$

and $W_B \in \mathbb{R}^{d \times D/2}$ as a random matrix with its entries drawn from $\mathcal{N}(0, \sigma^{-1})$. The resulting features are then used to approximate the kernel machine with the implicitly infinite dimensional feature space,

$$k(x, x') \approx \hat{\phi}(x)^T \hat{\phi}(x'). \tag{5}$$

**ArcCos kernel**    To yield a better connection to state-of-the-art neural networks we use the ArcCos kernel [2]

$$k(x, x') = \frac{1}{\pi} \|x\| \|x'\| J(\theta)$$

with $J(\theta) = (\sin \theta + (\pi - \theta) \cos \theta)$ and $\theta = \cos^{-1}(\frac{x \cdot x'}{\|x\| \|x'\|})$, the angle between $x$ and $x'$. The approximation is not based on a Fourier transform, but is given by

$$\hat{\phi}(x) = \sqrt{\frac{1}{D}} \max(0, W_B^T x) \tag{6}$$

with $W_B \in \mathbb{R}^{d \times D}$ being a random Gaussian matrix. This makes the approximated feature map of the ArcCos kernel closely related to ReLUs in deep neural networks.

**Neural network interpretation**    The approximated kernel features $\hat{\phi}(x)$ can be interpreted as the output of the hidden layer in a shallow neural network. To obtain the neural network interpretation, we rewrite Eq. 2 as the following

$$f(x) = W^T h(W_B^T x) + b, \tag{7}$$

with $W \in \mathbb{R}^{D \times c}$ with $c$ number of classes, and $b \in \mathbb{R}^c$. Here, the non-linearity $h$ corresponds to the obtained kernel approximation map. Now, we substitute $z = W_B^T x$ in Eqs. 4 and 6 yielding $h(z) = \sqrt{1/D} [\sin(z), \cos(z)]^T$ for the Gaussian kernel and $h(z) = \sqrt{1/D} \max(0, z)$ for the ArcCos kernel.

## 3.2 Adapting random kernel approximations

Having introduced the neural network interpretation of random features, the key difference between both methods is which parameters are trained. For the neural network, one optimizes the parameters in the bottom-layer and those in the upper layers jointly. For kernel machines, however, $W_B$ is fixed, i.e., the features are not adapted to the data. Hyper-parameters (such as $\sigma$ defining the bandwidth of the Gaussian kernel) are selected with cross-validation or heuristics [12, 6, 8]. Consequently, the basis is not directly adapted to the data, loss, and task at hand.

In our experiments, we consider the classification setting where for the given data $X \in \mathbb{R}^{n \times d}$ containing $n$ samples with $d$ input dimensions one seeks to predict the target labels $Y \in [0,1]^{n \times c}$ with a one-hot encoding for $c$ classes. We use accuracy as the performance measure and the multinomial-logistic loss as its surrogate. All our models have the same, generic form shown in Eq. 7. However, we use different types of basis functions to analyze varying degrees of adaptation. In particular, we study whether data-dependent basis functions improve over data-agnostic basis functions. On top of that, we examine how well label-informative, thus task-adapted basis functions can perform in contrast to the data-agnostic basis. Finally, we use end-to-end learning of all parameters to connect to neural networks.

**Random Basis - RB:** For data-agnostic kernel approximation, we use the current state-of-the-art of random features. Orthogonal random features [8, ORF] improve the convergence properties of the Gaussian kernel approximation over random Fourier features [30, 31]. Practically, we substitute $W_B$ with $1/\sigma \, G_B$, sample $G_B \in \mathbb{R}^{d \times D/2}$ from $\mathcal{N}(0,1)$ and orthogonalize the matrix as given in [8] to approximate the Gaussian kernel. The ArcCos kernel is applied as described above.

We also use these features as initialization of the following adaptive approaches. When adapting the Gaussian kernel we optimize $G_B$ while keeping the scale $1/\sigma$ fixed.

**Unsupervised Adapted Basis - UAB:** While the introduced random bases converge towards the true kernel with an increasing number of features, it is to be expected that an optimized approximation will yield a more compact representation. We address this by optimizing the sampled parameters $W_B$ w.r.t. the kernel approximation error (KAE):

$$\hat{L}(x, x') = \frac{1}{2}(k(x, x') - \hat{\phi}(x)^T \hat{\phi}(x'))^2 \tag{8}$$

This objective is kernel- and data-dependent, but agnostic to the classification task.

**Supervised Adapted Basis - SAB:** As an intermediate step between task-agnostic kernel approximations and end-to-end learning, we propose to use kernel target alignment [5] to inject label information. This is achieved by a target kernel function $k_Y$ with $k_Y(x, x') = +1$ if $x$ and $x'$ belong to the same class and $k_Y(x, x') = 0$ otherwise. We maximize the alignment between the approximated kernel $k$ and the target kernel $k_Y$ for a given data set $X$:

$$\hat{A}(X, k, k_Y) = \frac{\langle K, K_Y \rangle}{\sqrt{\langle K, K \rangle \langle K_Y, K_Y \rangle}} \tag{9}$$

with $\langle K_a, K_b \rangle = \sum_{i,j}^n k_a(x_i, x_j) k_b(x_i, x_j)$.

**Discriminatively Adapted Basis - DAB:** The previous approach uses label information, but is oblivious to the final classifier. On the other hand, a discriminatively adapted basis is trained jointly with the classifier to minimize the classification objective, i.e., $W_B, W, b$ are optimized at the same time. This is the end-to-end optimization performed in neural networks.

## 4 Experiments

In the following, we present the empirical results of our study, starting with a description the experimental setup. Then, we proceed to present the results of using data-dependent and task-dependent basis approximations. In the end, we bridge our analysis to deep learning and fast kernel machines.

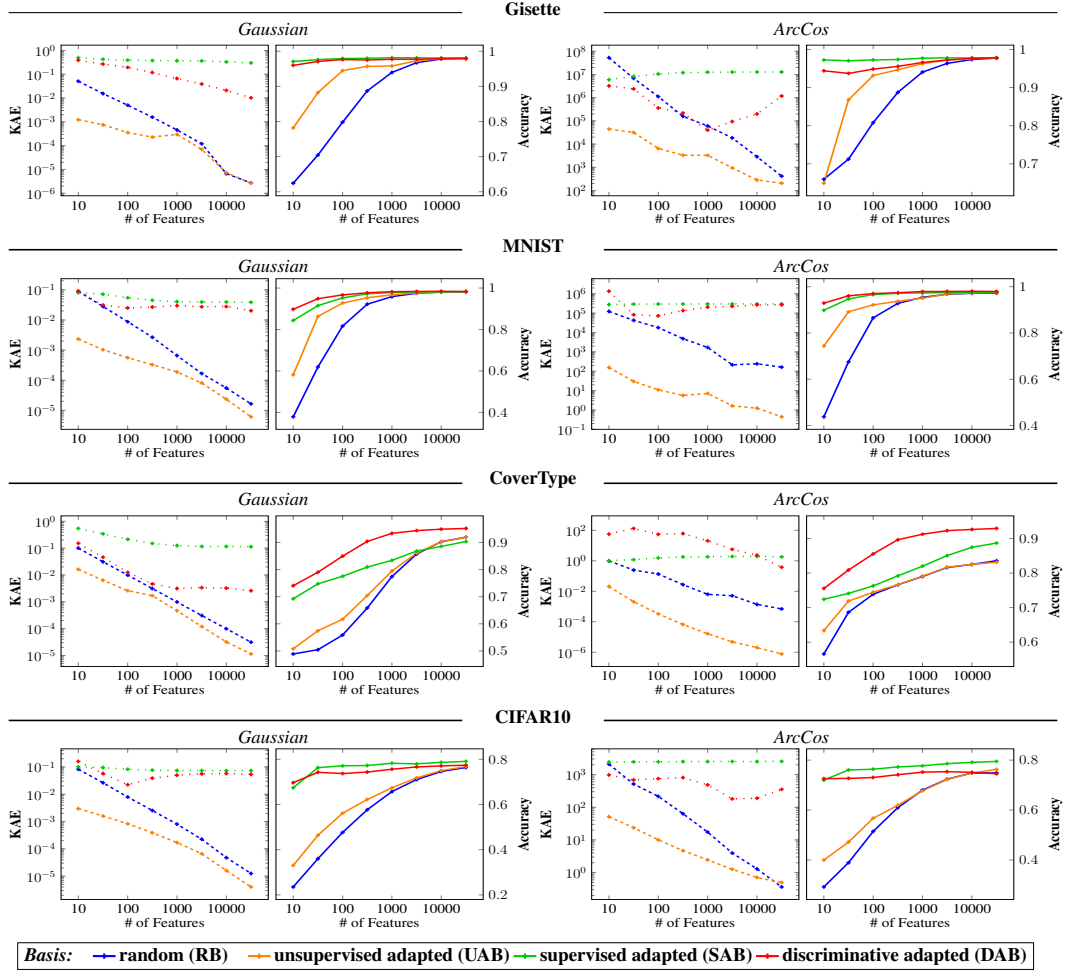

Figure 1: **Adapting bases.** The plots show the relationship between the number of features (X-Axis), the KAE in *logarithmic* spacing(**left, dashed lines**) and the classification error (**right, solid lines**). Typically, the KAE decreases with a higher number of features, while the accuracy increases. The KAE for SAB and DAB (orange and red dotted line) hints how much the adaptation deviates from its initialization (blue dashed line). Best viewed in digital and color.

## 4.1 Experimental setup

We used the following seven data sets for our study: Gisette [13], MNIST [21], CoverType [1], CIFAR10 features from [4], Adult [18], Letter [10], USPS [15]. The results for the last three can be found in the supplement. We center the data sets and scale them feature-wise into the range $[-1, +1]$. We use validation sets of size $1,000$ for Gisette, $10,000$ for MNIST, $50,000$ for CoverType, $5,000$ for CIFAR10, $3,560$ for Adult, $4,500$ for Letter, and $1,290$ for USPS. We repeat every test three times and report the mean over these trials.

**Optimization**   We train all models with mini-batch stochastic gradient descent. The batch size is $64$ and as update rule we use ADAM [17]. We use early-stopping where we stop when the respective loss on the validation set does not decrease for ten epochs. We use Keras [3], Scikit-learn [29], NumPy [33] and SciPy [16]. We set the hyper-parameter $\sigma$ for the Gaussian kernel heuristically according to [39, 8].

UAB ans SAB learning problems scale quadratically in the number of samples $n$. Therefore, to reduce memory requirements we optimize by sampling mini-batches from the kernel matrix. A batch for UAB consists of $64$ sample pairs $x$ and $x'$ as input and the respective value of the kernel function $k(x, x')$ as target value. Similarly for SAB, we sample $64$ data points as input and generate the

target kernel matrix as target value. For each training epoch we randomly generate $10,000$ training and $1,000$ validation batches, and, eventually, evaluate the performance on $1,000$ unseen, random batches.

## 4.2 Analysis

Tab. 1 gives an overview of the best performances achieved by each basis on each data set.

| Dataset | Gaussian | | | | ArcCos | | | |
|---------|------|------|------|------|------|------|------|------|
| | **RB** | **UAB** | **SAB** | **DAB** | **RB** | **UAB** | **SAB** | **DAB** |
| *Gisette* | 98.1 | 97.9 | 98.1 | 97.9 | 97.7 | 97.8 | 97.8 | 97.8 |
| *MNIST* | 98.2 | 98.2 | 98.3 | 98.3 | 97.2 | 97.4 | 97.7 | 97.9 |
| *CoverType* | 91.9 | 91.9 | 90.4 | 95.2 | 83.6 | 83.1 | 88.7 | 92.9 |
| *CIFAR10* | 76.4 | 76.8 | 79.0 | 77.3 | 74.9 | 76.3 | 79.4 | 75.3 |

Table 1: **Best accuracy** in % for different bases.

**Data-adapted kernel approximations**    First, we evaluate the effect of choosing a data-dependent basis (UAB) over a random basis (RB). In Fig. 1, we show the kernel approximation error (KAE) and the classification accuracy for a range from 10 to 30,000 features (in logarithmic scale). The first striking observation is that a data-dependent basis can approximate the kernel equally well with up to two orders of magnitude fewer features compared to the random baseline. This hold for both the Gaussian and the ArcCos kernel. However, the advantage diminishes as the number of features increases. When we relate the kernel approximation error to the accuracy, we observe that initially a decrease in KAE correlates well with an increase in accuracy. However, once the kernel is approximated sufficiently well, using more feature does not impact accuracy anymore.

We conclude that the choice between a random or data-dependent basis strongly depends on the application. When a short training procedure is required, optimizing the basis could be too costly. On the other hand, if the focus lies on fast inference, we argue to optimize the basis to obtain a compact representation. In settings with restricted resources, e.g., mobile devices, this can be a key advantage.

**Task-adapted kernels**    A key difference between kernel methods and neural networks originates from the training procedure. In kernel methods the feature representation is fixed while the classifier is optimized. In contrast, deep learning relies on end-to-end training such that the feature representation is tightly coupled to the classifier. Intuitively, this allows the representation to be tailor-made for the task at hand. Therefore, one would expect that this allows for an even more compact representation than the previously examined data-adapted basis.

In Sec. 3, we proposed a task-adapted kernel (SAB). Fig. 1 shows that the approach is comparable in terms of classification accuracy to discriminatively trained basis (DAB). Only for CoverType data set SAB performs significantly worse due to the limited model capabilities, which we will discuss below. Both task-adapted features improve significantly in accuracy compared to the random and data-adaptive kernel approximations.

**Transfer learning**    The beauty of kernel methods is, however, that a kernel function can be used across a wide range of tasks and consistently result in good performance. Therefore, in the next experiment, we investigate whether the resulting kernel retains this generalization capability when it is task-adapted. To investigate the influence of task-dependent information, we randomly separate the classes MNIST into two distinct subsets. The first task is to classify five randomly samples classes and their respective data points, while the second task is to do the same with the remaining classes. We train the previously presented model variants on task 1 and transfer their bases to task 2 where we only learn the classifier. The experiment is repeated with five different splits and the mean accuracy is reported.

Fig. 2 shows that on the transfer task, the random and the data-adapted bases RB and UAB approximately retain the accuracy achieved on task 1. The performance of the end-to-end trained basis DAB drops significantly, however, yields still a better performance than the default random basis. Surprisingly, the supervised basis SAB using kernel-target alignment retains its performance and achieves the highest accuracy on task 2. This shows that using label information can indeed be

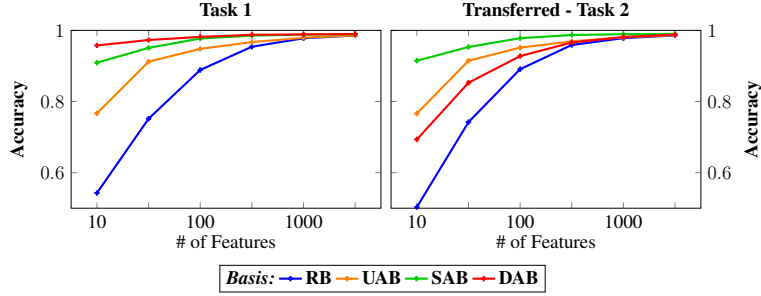

Figure 2: **Transfer learning.** We train to discriminate a random subset of 5 classes on the MNIST data set (left) and then transfer the basis function to a new task (right), i.e., train with the fixed basis from task 1 to classify between the remaining classes.

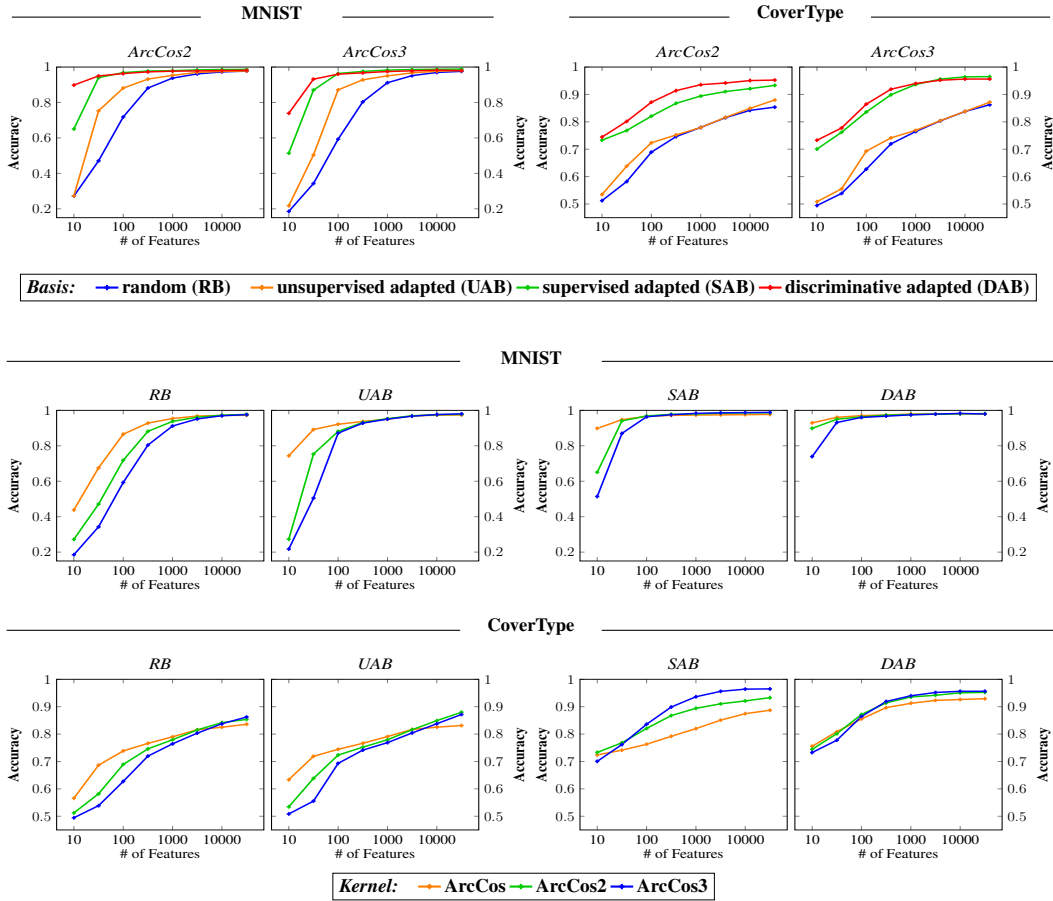

Figure 3: **Deep kernel machines.** The plots show the classification performance of the ArcCos-kernels with respect to the kernel (**first part**) and with respect to the number of layers (**second part**). Best viewed in digital and color.

exploited in order to improve the efficiency and performance of kernel approximations without having to sacrifice generalization. I.e., a target-driven kernel (SAB) can be an efficient and still general alternative to the universal Gaussian kernel.

**Deep kernel machines** We extend our analysis and draw a link to deep learning by adding two deep kernels [2]. As outlined in the aforementioned paper, stacking a Gaussian kernel is not useful instead we use ArcCos kernels that are related to deep learning as described below. Recall the ArcCos kernel from Eq. 3.1 as $k_1(x, x')$. Then the kernels ArcCos2 and ArcCos3 are defined by the inductive step

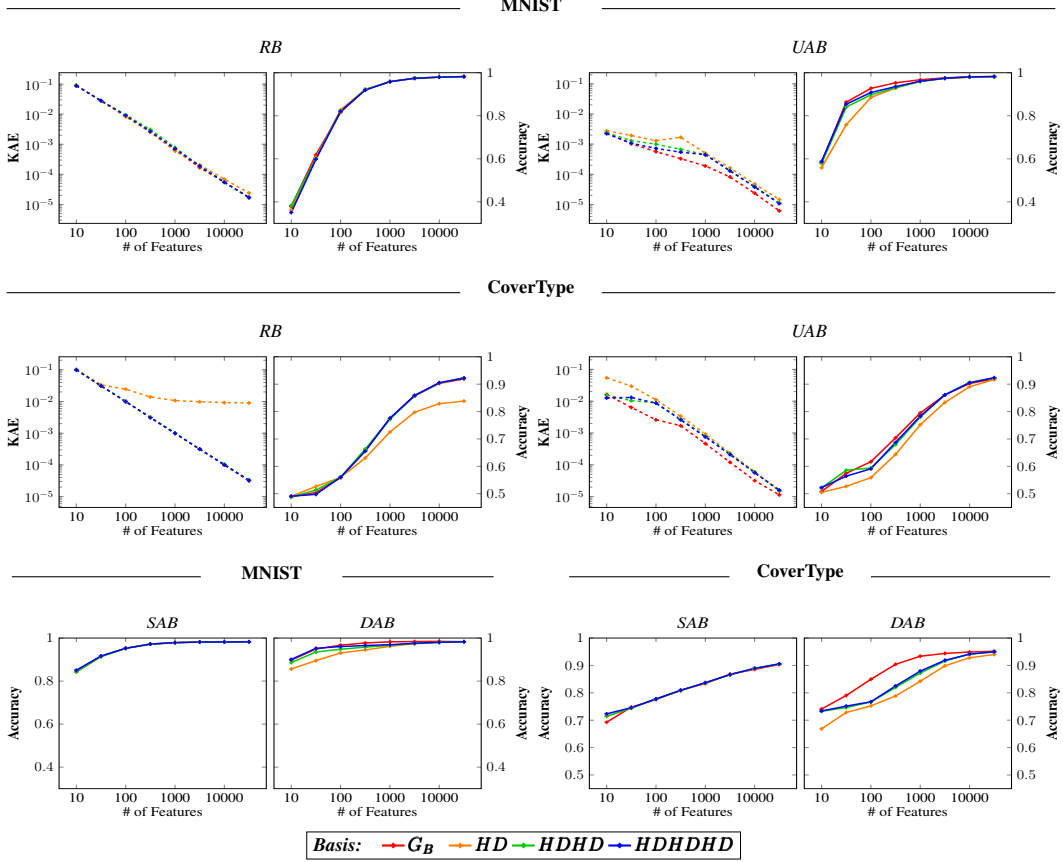

Figure 4: **Fast kernel machines.** The plots show how replacing the basis $G_B$ with an fast approximation influences the performance of a Gaussian kernel. I.e., $G_B$ is replaced by 1, 2, or 3 structured blocks $HD_i$. Fast approximations with 2 and 3 blocks might overlap with $G_B$. Best viewed in digital and color.

$k_{i+1}(x, x') = \frac{1}{\pi}[k_i(x, x)k_i(x', x')]^{-1/2}J(\theta_i)$ with $\theta_i = \cos^{-1}(k_i(x, x')[k_i(x, x)k_i(x', x')]^{-1/2})$. Similarly, the feature map of the ArcCos kernel is approximated by a one-layer neural network with the ReLU-activation function and a random weight matrix $W_B$

$$\hat{\phi}_{ArcCos}(x) = \hat{\phi}_B(x) = \sqrt{\frac{1}{D}}\max(0, W_B^T x), \qquad (10)$$

and the feature maps of the ArcCos2 and ArcCos3 kernels are then given by a 2- or 3-layer neural network with the ReLU-activations, i.e., $\hat{\phi}_{ArcCos2}(x) = \hat{\phi}_{B_1}(\hat{\phi}_{B_0}(x))$ and $\hat{\phi}_{ArcCos3}(x) = \hat{\phi}_{B_2}(\hat{\phi}_{B_1}(\hat{\phi}_{B_0}(x)))$. The training procedure for the ArcCos2 and ArcCos3 kernels remains identical to the training of the ArcCos kernel, i.e., the random matrices $W_{B_i}$ are simultaneously adapted. Only, now the basis consists of more than one layer, and, to remain comparable for a given number of features, we split these features evenly over two layers for a 2-layer kernel and over three layers for a 3-layer kernel.

In the following we describe our results on the MNIST and CoverType data sets. We observed that the so far described relationship between the cases RB, UAB, SAB, DAB also generalizes to deep models (see Fig. 3, first part, and Fig. 7 in the supplement). I.e., UAB approximates the true kernel function up to several magnitudes better than RB and leads to a better resulting classification performance. Furthermore, SAB and DAB perform similarly well and clearly outperform the task-agnostic bases RB and UAB.

We now compare the results across the ArcCos-kernels. Consider the third row of Fig. 3, which depicts the performance of RB and UAB on the CoverType data set. For a limited number of features, i.e., less than $3,000$, the deeper kernels perform worse than the shallow ones. Only given enough capacity the deep kernels are able to perform as good as or better than the single-layer bases. On the

other hand for the CoverType data set, task related bases, i.e., SAB and DAB, benefit significantly from a deeper structure and are thus more efficient. Comparing SAB with DAB, for the ArcCos kernel with only one layer SAB leads to worse results than DAB. Given two layers the gap diminishes and vanishes with three layers (see Fig. 3). This suggests that for this data set the evaluated shallow models are not expressive enough to extract the task-related kernel information.

**Fast kernel machines**   By using structured matrices one can speed up approximated kernel machines [20, 8]. We will now investigate how this important technique influences the presented basis schemes. The approximation is achieved by replacing random Gaussian matrices with an approximation composed of diagonal and structured Hadamard matrices. The advantage of these matrix types is that they allow for low storage costs as fast multiplications. Recall that the input dimension is $d$ and the number of features is $D$. By using the fast Hadamard-transform these algorithms only need to store $O(D)$ instead of $O(dD)$ parameters and the kernel approximation can be computed in $O(D \log d)$ rather than $O(Dd)$.

We use the approximation from [8] and replace the random Gaussian matrix $W_B = 1/\sigma \, G_B$ in Eq. 4 with a chain of random, structured blocks $W_B \approx 1/\sigma \, HD_1 \, \ldots \, HD_i$. Each block $HD_i$ consists of a diagonal matrix $D_i$ with entries sampled from the Rademacher distribution and a Hadamard matrix $H$. More blocks lead to a better approximation, but consequently require more computation. We found that the optimization is slightly more unstable and therefore stop early only after 20 epochs without improvement. When adapting a basis we will only modify the diagonal matrices.

We re-conducted our previous experiments for the Gaussian kernel on the MNIST and CoverType data sets (Fig. 4). In the first place one can notice that in most cases the approximation exhibits no decline in performance and that it is a viable alternative for all basis adaption schemes. Two major exceptions are the following. Consider first the left part of the second row which depicts a approximated, random kernel machine (RB). The convergence of the kernel approximation stalls when using a random basis with only one block. As a result the classification performance drops drastically. This is not the case when the basis is adapted unsupervised, which is given in the right part of the second row. Here one cannot notice a major difference between one or more blocks. This means that for fast kernel machines an unsupervised adaption can lead to a more effective model utilization, which is crucial for resource aware settings. Furthermore, a discriminatively trained basis, i.e., a neural network, can be effected similarly from this re-parameterization (see Fig. 4, bottom row). Here an order of magnitude more features is needed to achieve the same accuracy compared to an exact representation, regardless how many blocks are used. In contrast, when adapting the kernel in a supervised fashion no decline in performance is noticeable. This shows that this procedure uses parameters very efficiently.

## 5   Conclusions

Our analysis shows how random and adaptive bases affect the quality of learning. For random features this comes with the need for a large number of features and suggests that two issues severely limit approximated kernel machines: the basis being (1) agnostic to the data distribution and (2) agnostic to the task. We have found that data-dependent optimization of the kernel approximation consistently results in a more compact representation for a given kernel approximation error. Moreover, task-adapted features could further improve upon this. Even with fast, structured matrices, the adaptive features allow to further reduce the number of required parameters. This presents a promising strategy when a fast and computationally cheap inference is required, e.g., on mobile device.

Beyond that, we have evaluated the generalization capabilities of the adapted variants on a transfer learning task. Remarkably, all adapted bases outperform the random baseline here. We have found that the kernel-task alignment works particularly well in this setting, having almost the same performance on the transfer task as the target task. At the junction of kernel methods and deep learning, this shows that incorporating label information can indeed be beneficial for performance without having to sacrifice generalization capability. Investigating this in more detail appears to be highly promising and suggests the path for future work.

**Acknowledgments**

MA, KS, KRM, and FS acknowledge support by the Federal Ministry of Education and Research (BMBF) under 01IS14013A. PJK has received funding from the European Union's Horizon 2020 research and innovation program under the Marie Sklodowska-Curie grant agreement NO 657679. KRM further acknowledges partial funding by the Institute for Information & Communications Technology Promotion (IITP) grant funded by the Korea government (No. 2017-0-00451), BK21 and by DFG. FS is partially supported by NSF IIS-1065243, 1451412, 1513966/1632803, 1208500, CCF-1139148, a Google Research Award, an Alfred. P. Sloan Research Fellowship and ARO# W911NF-12-1-0241 and W911NF-15-1-0484. This work was supported by NVIDIA with a hardware donation.

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
