[Supplementary Material]

# 6 Supplement

## 6.1 Additional empirical evidence

In Table 2 and Fig. 5 we show the results of our main experiment for three additional data sets. Please find the analysis in the main text.

| Dataset | Gaussian | | | | ArcCos | | | |
|---|---|---|---|---|---|---|---|---|
| | RB | UAB | SAB | DAB | RB | UAB | SAB | DAB |
| Adult | 85.1 | 85.0 | 84.8 | 85.1 | 85.0 | 85.1 | 84.9 | 85.0 |
| Letter | 96.1 | 96.1 | 97.1 | 97.6 | 95.3 | 95.3 | 90.0 | 98.7 |
| USPS | 95.1 | 95.0 | 94.5 | 95.3 | 94.3 | 94.4 | 92.0 | 95.1 |

Table 2: **Best accuracy** in % for different bases.

Figure 5: **Adapting bases.** The plots show the relationship between the number of features (X-Axis), the KAE in *logarithmic* spacing(**left, dashed lines**) and the classification error (**right, solid lines**). Typically, the KAE decreases with a higher number of features, while the accuracy increases The KAE for SAB and DAB (orange and red dotted line) hints how much the adaptation deviates from its initialization (blue dashed line). Best viewed in digital and color.

## 6.2 Deep kernel machines

Fig. 6 and Fig. 7 depict in more detail how the kernels ArcCos2 and ArcCos3 perform on the MNIST and the CoverType data set. Please find the analysis in the main text.

Figure 6: **Deep kernel machines.** The performance of the ArcCos-kernels with 1-, 2-, and 3-layer models. The KAE is given in dashed lines and the accuracy in solid lines. Best viewed in digital and color.

Figure 7: **Deep kernel machines.** The plots show the relationship between the number of features (X-Axis), the KAE in *logarithmic* spacing(**left, dashed lines**) and the classification error (**right, solid lines**). Typically, the KAE decreases with a higher number of features, while the accuracy increases The KAE for SAB and DAB (orange and red dotted line) hints how much the adaptation deviates from its initialization (blue dashed line). Best viewed in digital and color.

## 6.3 Optimization

Figure 8: **Optimization:** Comparison of the optimization duration (**solid**) in epochs of the cos-sin and the ReLu non-linearity given a varying number of features on the MNIST benchmark. For reference the obtained accuracies are plotted as **dashed** lines.

In general, the periodic nature of the sine and cosine function imposes limitations to applicability in neural networks. When scaled and initialized properly we found that the activation function in Eq. 4 can be optimized and reaches similar performances as a ReLU-powered neural network. From Fig. 8 one can see that the *sin-cos* activation function needs only a reasonable amount of epochs more to converge than with the ReLU activation. Overall, the training time of the different bases relate as follows. With respect to the classification performance SAB and DAB are considerably faster than RB and UAB. This holds mainly because one can train a much smaller basis while reaching the same performance. With regard to the number of features all methods expose a linear increase in training time. This is caused by the chosen learning procedure. Given the same number of features, methods without kernel adaption, i.e., RB and DAB, are up to a magnitude faster than the others. Further, training using a RB can be up to a magnitude faster than DAB. Note that we did not tune nor implement the kernel adaption to be fast, but to give high accuracy.