[Reviews · NeurIPS 2017]

Reviewer 1



he paper presents an experimental evaluation of different kernel methods using random bases applied to classification tasks. Despite the focus of the paper on empirical evaluation, the paper contributes new insights that relate kernel approximation methods to neural networks. In particular the paper shows that a kernel classifier using random Fourier features (RFF) for kernel approximation can be cast as a shallow neural network. The experimental results are interesting and show that allowing the learning of the RFF's random Gaussian matrix W_B produce it is possible to obtain high accuracy with several orders of magnitude less features. This said, I think that the contribution of the paper is still short of what is expected for the conference. The introduction creates the false expectation that the paper will provide insights on the differences/similarities between kernel methods and deep learning architectures (from the introduction "Understanding these differences in efficiency between approximated kernel methods and deep architecture is crucial to use them optimally in practice"). However, the experiments just compare an approximate kernel method against adaptive version of the same method, which are equivalent to a shallow neural network. The extension of the results and experimentation to deep learning architectures would make a very good contribution.

Reviewer 2



The authors conduct an interesting empirical study on the use of random bases in kernel methods and its connection and interpretation to neural networks. Up to my knowledge the ArcCos kernel in connection to the ReLU neural networks interpretation in eq (5) is new and an interesting result. My suggestion is to stress more this novel contribution, also in the abstract. The title suggests that the paper is only focusing on an empirical study, but this eq (5) looks novel and interesting to me. In section 3.1 the authors could somewhat better specify which parts are new versus existing. The connection between neural networks and kernel methods in eqs (2)(6) has been discussed in previous work: in the book J.A.K. Suykens, T. Van Gestel, J. De Brabanter, B. De Moor, J. Vandewalle, Least Squares Support Vector Machines, World Scientific, Singapore, 2002 neural networks interpretations were given both to the primal representation (2) and the dual representation with the kernel function, together with approximations to the feature map, by Nystrom approximation on a subset. The paper Suykens J.A.K., Vandewalle J., Training multilayer perceptron classifiers based on a modified support vector method, IEEE Transactions on Neural Networks, vol.10, no.4, Jul. 1999, pp.907-911. explicity took the feature map equal to the hidden layer of an MLP neural network. The experiments contain an interesting comparison between RB, UAB, SAB, DAB, illustrating the effect of the number of features. However, shouldn't the number of features be considered as a tuning parameter? It would be good to pay more attention to the definition of the tuning parameters in the different methods, and how they are selected. In Table 1 only the best accuracy is given. Also to mean accuracy (and standard deviation) would be interesting to know here.

Reviewer 3



Summary: The authors provided an empirical study contrasting neural networks and kernel methods, with a focus on how random and adaptive schemes would make efficient use of features in order to improve quality of learning, at four levels of abstraction: data-agnostic random basis (baseline kernel machines with traditional random features), unsupervised data-adaptive basis for better approximation of kernel function, supervised data-label-adaptive basis by kernel target alignment, discriminatively adaptive basis (neural nets). The paper concluded with several suggestions and caveats for efficient use of random features in practice. Comments: - 1 - Line 123, especially for sake of comparing UAB case where the underlying assumption is that using the true kernel function k in prediction yields the "best" performance so that UAB tries to approximate it, I would suggest testing in experiments a baseline model that utilizes the true kernel function k in prediction. Also this would suggest, for example in Fig. 1 at which point of the KAE curve the accuracy is sufficiently good (despite many theoretical results available). - 2 - The four datasets chosen in the paper certainly demonstrate proof for conclusions finally drawn. However, in order to support those conclusions to a definitively convincing extent, more datasets should be needed. For example, the performance scores in Tab. 1 do not seem to be too significantly different marginally for each task. And reasons for inconsistent behaviors across different tasks (CoverType vs others in Fig. 1 for instance, Line 222) are not well explained through empirical exploration. - 3 - It is not clear how big the difference can be between the four adaptive schemes in terms of training time, which can be crucial in practice. In addition, as the number of (approximate) features D increases, how does the training time increase accordingly in general for each scheme empirically? It would be interesting to also report this.